# The Types of Polysaccharide Coatings and Their Mixtures as a Factor Affecting the Stability of Bioactive Compounds and Health-Promoting Properties Expressed as the Ability to Inhibit the α-Amylase and α-Glucosidase of Chokeberry Extracts in the Microencapsulation Process

**DOI:** 10.3390/foods10091994

**Published:** 2021-08-25

**Authors:** Kamil Haładyn, Karolina Tkacz, Aneta Wojdyło, Paulina Nowicka

**Affiliations:** Department of Fruit, Vegetable and Nutraceutical Plant Technology, The Faculty of Biotechnology and Food Science, Wrocław University of Environmental and Life Sciences, 37 Chełmońskiego Street, 51-630 Wrocław, Poland; kamil.haladyn@gmail.com (K.H.); karolina.tkacz@upwr.edu.pl (K.T.); aneta.wojdylo@upwr.edu.pl (A.W.)

**Keywords:** microencapsulation, black chokeberry, microsphere, chitosan, sodium alginate, guar gum, polyphenolic compounds

## Abstract

This study aimed to evaluate the feasibility of microencapsulating chokeberry extract by extrusion, and assess the effects of the selected carrier substance on the contents of polyphenolic compounds, antioxidant activity, color of microspheres, and ability of microspheres to inhibit α-amylase and α-glucosidase, after 14 and 28 days of storage. The results showed that appropriate selection of the polysaccharide coating is of great importance for the proper course of the microencapsulation process, the polyphenolic content of chokeberry capsules, and their antioxidant and antidiabetic properties. The addition of guar gum to a sodium alginate solution significantly increased the stability of polyphenolic compounds in microspheres during storage, whereas the addition of chitosan had a significantly negative effect on the stability of polyphenols. The coating variant composed of sodium alginate and guar gum was also found to be the most favorable for the preservation of the antioxidant activity of the capsules. On the other hand, capsules composed of sodium alginate, guar gum, and chitosan showed the best antidiabetic properties, which is related to these tricomponent microspheres having the best α-glucosidase inhibition.

## 1. Introduction

Growing awareness of the impact of nutrition on human health has contributed to a change in the eating habits of society [1]. In recent years, raw materials of plant origin have been playing a major role in the daily diet. In particular, fruits and vegetables, the recommended intake of which is shown to have potential health-promoting effects on the body, constitute an important part of the diet [2]. Fruits are a valuable source of bioactive compounds [3]. These naturally occurring substances are nutritional or non-nutritional in nature and influence the physiological functions of the body. Nutrients act as a source of energy and building material, and thus are essential for the proper functioning of the body, whereas non-nutritional compounds play an important role in maintaining human health, although they are not necessary for the functioning of specific systems [1]. Consumption of fruits may also prevent the onset of some diet-related diseases, such as cardiovascular disorders, type II diabetes, gastrointestinal disorders, and cancers [4,5,6,7]. The valuable health effects of fruits are associated with the presence of secondary plant metabolites, which include polyphenolic compounds, terpenoids, and alkaloids [8,9]. Among fruits, chokeberry is a rich source of polyphenols [10].

Aronia melanocarpa L. is a deciduous shrub of the rose family (Rosaceae) and is native to North America. Its fruits are dark in color and have a characteristic astringent taste, which is caused by the presence of polyphenolic compounds [11,12,13]. These compounds include the polymers of procyanidins, anthocyanins, phenolic acids, and flavonols, and monomers of flavan-3-ol [14]. Similarly to other plant-derived raw materials, polyphenols present in chokeberry fruits exhibit a number of beneficial properties [15]. The occurrence of polyphenolic compounds with antioxidant properties protects the cells from oxidative stress caused by the presence of reactive oxygen species [16,17]. Unfortunately, plant-derived polyphenols are very sensitive to changes in environmental factors, such as temperature, light, and the presence of oxygen, and so they are easily degraded and lose their valuable properties. Degradation of polyphenols also occurs in the human digestive system when the compounds are supplied to the body. However, this problem may be overcome by the microencapsulation process [18].

Encapsulation is the process of coating a core substance with a coating material, which results in the formation of a capsule [19,20]. Depending on size, capsules are distinguished into microcapsules and nanocapsules, among others. The main purpose of microencapsulation is to protect biologically active compounds, such as natural dyes (e.g., anthocyanins), vitamins, and polyphenols, from degradation. Additionally, this process allows the controlled release of core substances by delaying their absorption in the gastrointestinal tract and preventing their degradation in the initial digestion process [21]. Hermetization can be done using different methods. One of them is the extrusion technique, which results in the formation of a gel-like polymer capsule. The resulting polymer solution is cured by contact with a solidifying liquid [22,23]. The capsules thus obtained can vary in size from 400 to 2000 µm. The use of these encapsulation processes allows lowering the manufacturing costs, eliminating the use of organic solvents, and avoiding high temperatures. Hence, the extrusion method can be safely used to encapsulate bioactive compounds without thermal degradation [21,23,24].

Many polymers can be used for encapsulation. Sodium alginate is mainly used for microencapsulating a compound by extrusion. This polysaccharide consists of multiple α-D-mannuronic and α-L-guluronic acid units, arranged linearly and connected by β-1,4-glycosidic bonds [25]. Sodium alginate is resistant to acidic pH; however, in the alkaline environment, the microspheres made using it swell and disintegrate [26], which is a major drawback and may limit the use of this polymer in microencapsulation. Chitosan is a polymer derived from the deacetylation of chitin, a substance found naturally in the cell walls of selected fungal species and in the skeletons of crustaceans and insects [27,28]. This polymer can bind cholesterol in the gastrointestinal tract, and exhibits antioxidant and antibacterial properties, and hence has high health-promoting potential [28,29]. In turn, due to the functional groups present in the polysaccharide chain, this polymer allows the controlled release of encapsulated material [30]. However, the use of chitosan is limited by its solubility, which occurs only in acidic environments [31]. Guar gum, on the other hand, is a polymer produced from the seeds of Cyamopsis tetragonoloba [32]. It is composed of multiple molecules of D-mannose and D-galactose, and therefore, has a high molecular weight and high viscosity [33]. An advantage of guar gum is its resistance to the action of digestive enzymes and to degradation in the initial part of the digestive system. Due to these properties, this polymer can be successfully used in the microencapsulation process [31].

Studies have shown the effects of different microencapsulation techniques on the contents of polyphenolic compounds and the antioxidant activity of the studied material, but these publications are related to freeze-drying. It seems that no study so far has focused on the formation of microgel capsules containing isolated bioactive compounds of chokeberry using the extrusion technique. Moreover, the available literature lacks information on the influences of plant polymers on the behaviors of bioactive compounds during microsphere storage. Taking these into account, this study was performed to evaluate the feasibility of microencapsulating chokeberry extract by extrusion and to assess the effect of the selected carrier substance on the content of polyphenolic compounds, antioxidant activity, and color of microspheres, and the ability of microspheres to inhibit α-amylase and α-glucosidase, after 14 and 28 days of storage.

It was assumed that the results of this work would allow the selection of appropriate polysaccharide material that can retain the stability of bioactive compounds during storage. Additionally, it was hypothesized that the obtained results will reveal the biological and health-promoting potential of the produced microspheres and will contribute to determining the potential application of microspheres in food fortification.

## 2. Materials and Methods

### 2.1. Reagents

The following polysaccharides were used to prepare coating materials: low-molecular-weight chitosan (Aldrich-Sigma, Saint Louis, MO, USA), sodium alginate (Aldrich-Sigma, Shanghai, China), and guar gum (Agnex, Białystok, Poland). Calcium chloride dihydrate (Sigma-Aldrich) was used for the solidification of capsules.

### 2.2. Extraction of Chokeberry Fruits

Chokeberry fruits were obtained in October 2019 from orchards in Trzebnica, which is located near Wroclaw (Poland). The collected fruits were stored in a freezer at 20 °C until the start of the experiments. Shortly before the analyses, they were thawed, crushed, and heated to 40 °C (Thermomix, Wuppertal, Germany), and treated with 0.14% (*v*/*v*) Pectinex AFP L-4 enzyme preparation (Novozymes, Bagsvaerd, Denmark). After this procedure, the juice was squeezed from the material on a hydraulic press and applied to a column packed with Amberlite^®^ XAD-16 resin to remove sugars and high-molecular-weight compounds. The bioactive compounds absorbed on the bed were eluted using 80% (*v*/*v*) ethanol, and the resulting extract was transferred to a rotary evaporator (Hei-VAP Expert Control; Heidolph, Schwabach, Germany) for ethanol removal. The prepared product was then concentrated to form a concentrate of polyphenolic compounds. A part of the extract was lyophilized (lyophilizer OE-950 Labor; MIM, Budapest, Hungary), and chokeberry powder was prepared.

### 2.3. Microencapsulation Procedure

To obtain microcapsules, sodium alginate (Alg)—1 g, guar gum (Gum)—0.2 g, and chitosan (Chit)—0.2 g were mixed and the mixture was dissolved in 100 mL of distilled water at pH 5.0. The solution was then stirred for 180 min with a magnetic stirrer and homogenized. To determine the potential effect of the encapsulating substance on the contents of bioactive compounds, three additional solutions were made in which 0.2 g of chitosan, 0.2 g of guar gum, and 0.2 g of chitosan and guar gum were eliminated sequentially (leaving only 1.0 g of sodium alginate). The primary substance surrounding the bioactive compounds was a ternary coating. Sodium alginate could not be removed as it was essential for the microencapsulation method used and for the formation of capsules.

The powdered formulation (0.8 g) containing purified and isolated bioactive compounds was dissolved in four types of biopolymer solutions (100 mL). The resulting solutions were then mixed and homogenized (10,000 rpm). The entire process was performed in duplicate, so that half of the material was filtered to eliminate particulate matter and the other half was left unfiltered. All these operations were performed identically for the aronia concentrate. The amount of material dissolved in 100 mL of biopolymer solution was 2 mL. All the amounts used in the present experiment were selected based on an empirical analysis. The prepared mixtures were ready for use in the microencapsulation process. A B-390 microencapsulator (BÜCHI, Flawil, Switzerland) was used to produce microspheres under appropriate process parameters (flow frequency, vibration amplitude, and voltage). The pumped stream of the biopolymer solution was directed to a vessel containing 1.5% calcium chloride solution, which was located on a magnetic stirrer plate. The residence time of the capsules in this solidification bath was 20 min. The capsules thus obtained were used for further analysis.

### 2.4. Analysis of Phenolic Compounds

The polyphenolic compounds were identified and quantified by liquid chromatography quadrupole time of flight mass spectrometry (LC/MS-Q/TOF) and ultraperformance liquid chromatography (UPLC)-PDA after a prior extraction step. For this purpose, approximately 1.5 g of microspheres was weighed and mixed with 5 mL of aqueous methyl alcohol solution (30%, *v*/*v*) with 2% ascorbic acid and 1% acetic acid. The samples were subjected to ultrasound for 25 min and then shaken (450 rpm, 60 min). The whole process was repeated after storing the samples for 24 h at 4 °C. The obtained extracts were filtered through a hydrophilic membrane hydrophilic polytetrafluoroethylene (PTFE, 0.20 µm; Milex Samplicity Filter; Merck, Drmstadt, Germany), and the filtrate was used for further analyses.

The polyphenolic compounds were analyzed as described by Wojdylo et al. [34] using an Acquity UPLC system (Waters Corporation, Milford, MA, USA). The chromatograph was equipped with a photodiode array detector and a binary pump system containing a solvent manager. An AQUITY BEH C18 column (2.1 × 100 mm, 1.7 µm; Waters Corporation, Milford, MA, USA) was thermostated at 30 °C. The injection volume was 5 µL, and the flow rate was set to 0.420 mL/min. The polyphenolic compounds were separated in a gradient system consisting of phase A (4.5% formic acid) and phase B (100% acetonitrile). Elution was as follows: 0–10 min-linear gradient from 1 to 10% B; after that, 10–15 min-linear gradient from 10 to 17% B; and finally, 100% B from minute 15 to minute 18 for column washing, and reconditioning for next 4 min. Acetonitrile (100%) was used as a strong wash solvent and 10% of acetonitrile solution as a weak wash solvent. Detection of compounds was carried out at the following wavelengths (λ): flavon-3-ols, 280 nm; phenolic acids, 320 nm; flavonols, 360 nm; and anthocyanins, 520 nm. The results were determined using Empower 3.0 software and are presented in mg/100 g of product.

### 2.5. Determination of Polymeric Procyanidin Content by the UPLC-PDA-FL

The contents of procyanidin polymers were determined by fluoroglucinolysis, as described by Wojdylo et al. [34]. Briefly, approximately 500 mg of microspheres was weighed and lyophilized, and then 0.8 mL of fluoroglucinol and 0.4 mL of methanol were added. The samples were thermostated at 55 °C for 30 min and transferred to cooling blocks with 0.6 mL of sodium acetate solution. The material was centrifuged (14,000× *g*, 5 min), and 0.6 mL of supernatant was collected. The supernatant was treated with 0.6 mL of sodium acetate and centrifuged again. After centrifugation, the liquid layer was poured off, and the precipitate was used for further analysis. Procyanidins were quantified in an Acquity UPLC system (Waters Corporation, Milford, MA, USA) equipped with a binary solvent manager and a fluorescence detector (FL). Analysis was performed using an Acquity BEH (Ethylene Bridged Hybrid) Shield C18 column (2.1 × 50 mm, 1.7 µm) thermostated at 15 °C. The injection volume was 5 µL. Separation was carried out in a gradient system consisting of solvent A (2.5% acetic acid) and solvent B (100% acetonitrile). The flow rate was set at 0.45 mL/min. Measurement was performed at an excitation wavelength of 270 nm and an emission wavelength of 360 nm. The obtained results were processed using Empower 3.0 and are presented in mg/100 g.

### 2.6. Determinations of Antioxidant Aapacity (ABTS, FRAP, and ORAC Methods) and Biological Activity

The same sample preparation procedure was used for all methods. Briefly, to approximately 1.5 g of material, 7 mL of 80% methanol solution containing a small amount of HCl (1 mL/L) was added. The rest of the procedure was the same as that described for the extraction of polyphenolic compounds for determination by UPLC.

The antioxidant activity of the obtained microspheres was determined by their ability to inactivate ABTS radical cation (ABTS method), reduce ferric ions (FRAP method), and adsorb free radicals (free radical absorption capacity (ORAC) method), as described by Re et al. [35], Benzie and Strain [36], and Ou et al. [37], respectively. The results were expressed as mmol Trolox/100 g product. 

The ability to inhibit α-amylase and α-glucosidase was tested using the procedure described by Nowicka et al. [38]. Briefly, the inhibitory activity of the samples against α-amylase enzyme was determined by measuring their absorbance at 600 nm after the addition of a coloring reagent. Absorbance was based on the reaction of the iodine in potassium iodide with the sugars remaining in the samples after enzymatic hydrolysis. 

On the other hand, the α-glucosidase-inhibiting ability of each sample was determined by measuring the absorbance at 405 nm. For this, biologically active compounds extracted from the microspheres were dissolved in phosphate buffer and mixed with a phosphate buffer containing p-nitro-phenyl-α-D-glucopyranoside. The resulting samples were incubated at 37 °C for 15 min, and their absorbances were measured at 405 nm on a UV-240 PC spectrophotometer (Shimadzu, Kyoto, Japan).

### 2.7. Color Measurement in the CIE L*a*b System

The color of the obtained microspheres was analyzed using a CM-700d spectrophotometer (Konica Minolta Inc., Tokyo, Japan). This measurement was performed on an evenly distributed layer of capsules on a plastic Petri dish. The obtained results are presented in CIE L*a*b color space using the parameters L*, a*, and b*.

### 2.8. Optical Microscopy Analysis

The microspheres were imaged and measured using an Axiolab 5 microscope (Zeiss, Jena, Germany) integrated with an Axiocam 208 color microscope camera. The material was viewed at a magnification of ×5 for the analysis.

### 2.9. Statistical Analysis

Statistical analysis of the results was performed using Statistica 13.1 software (StatSoft, Krakow, Poland) based on a multifactor analysis of variance (*p* ≤ 0.05) and Duncan’s test. The results are presented–in the case of antioxidant activity as mean values (*n* = 3) ± standard deviation; in the case of content of phenolic compounds as mean values (*n* = 3); in the case of inhibitory effect as IC_50_ (*n* = 9).

## 3. Results

### 3.1. Analysis of Phenolic Compounds and Polymeric Procyanidin Contents

#### 3.1.1. Total Polyphenolic Compounds in Microspheres

Table 1 shows the contents of polyphenolic compounds determined in microspheres containing chokeberry concentrate and powder. Among the unstored capsules, the highest total polyphenol content was recorded for the variant containing sodium alginate, chitosan, and guar gum (Alg:Chit:Gum) (microspheres with concentrate: 108.29 mg/100 g product; microspheres with powder: 158.08 mg/100 g product). On the other hand, the lowest concentration of polyphenolic compounds was recorded in the single-component microcapsules constructed with sodium alginate (Alg) in the case of concentrate (53.46 mg/100 g product) and in the microcapsules constructed with sodium alginate and chitosan in the case of powder (53.01 mg/100 g product). The total contents of polyphenolic compounds in the microspheres obtained from previously filtered material were found to be different. For concentrate, the highest polyphenol content was observed in capsules made from a combination of sodium alginate and chitosan (Alg:Chit), whereas the lowest concentration of the compounds was found in alginate microspheres. In the case chokeberry powder, the microspheres composed of sodium alginate and guar gum (Alg:Gum) had the highest content of polyphenolic compounds (89.32 mg/100 g product), whereas the Alg variant had the lowest content of polyphenolic compounds (36.2 mg/100 g product). The application of the filtration process resulted in a significant decrease in the content of bioactive compounds in unstored microspheres. Depending on the microcapsule variant used, a decrease in the concentration of encapsulated material was observed, ranging from 15% for the Alg:Chit coating to 47% for the Alg:Gum blend, in comparison to the unfiltered counterparts. For the gel powder microspheres, the greatest difference in the content was observed for the capsule composed of the ternary coating, in which only 25% of the total polyphenol content was attributable to the unfiltered variant. Similarly to the encapsulation of chokeberry concentrate, the lowest decrease in the concentration of polyphenolic compounds in the microspheres composed of filtered biopolymer solutions was recorded for the Alg:Chit variant, which was about 1% for this combination. The significant reduction in the polyphenol content among the microencapsulates made from the filtered biopolymer solution containing guar gum was presumably related to the very high molecular weight of the polymer. Most probably, this high-molecular-weight polysaccharide was deposited on the semipermeable filter membrane, which in turn contributed to decreases in the filtration efficiency and retention of polyphenolic compounds on the permeable material.

#### 3.1.2. Polyphenolic Profile of Chokeberry Microspheres

The microspheres made from the unfiltered solution containing concentrate and guar gum (Alg:Chit:Gum and Alg:Gum) had the highest concentrations of polyphenolic compounds (108.29 and 102.13 mg/100 g product, respectively). In both the microcapsule variants, procyanidin polymers accounted for the highest proportion, constituting about 50% of the total polyphenols, while the second largest group of compounds in these variants was anthocyanins, constituting about 30%. The total value of anthocyanins in the two variants was 32.9 and 30.58 mg/100 g product, respectively. The next largest group of polyphenols found in the microspheres was phenolic acids. Their concentration in the Alg:Chit:Gum and Alg:Gum variants was recorded at 11% (12.06 mg/100 g) and 13% (12.93 mg/100 g), respectively. The least abundant were flavan-3-ol mono and dimers (4%). In contrast, the content of flavonols was 4.99 mg/100 g in the ternary microspheres and 4.60 mg/100 g in the Alg:Gum variant, which corresponded to 5% and 4% of all compounds present in the capsule. The profile of bioactive compounds differed in microspheres composed of sodium alginate and chitosan and in single-component alginate capsule. In both cases, the dominant group of compounds was anthocyanins, the content of which was determined at about 48%.

Regarding microcapsules with chokeberry powder, the proportion of each polyphenol group varied depending on the variant. In the Alg:Chit and Alg variants, the content of anthocyanins was about 45%, which constituted the highest proportion of total polyphenols. In the microcapsules containing sodium alginate and chitosan (without filtration), the second most abundant group of compounds was proanthocyanidin polymers, which had a total share of about 20% (28.96 mg/100 g product). On the other hand, the single-component alginate coating variant was characterized by a higher content of phenolic acids than proanthocyanidin polymers, the percentage of which in the total amount of polyphenols was 25% (22.84 mg/100 g product) and 13% (11.81 mg/100 g product), respectively. In the remaining microspheres, the main group of polyphenolic compounds was polymers of proanthocyanidins, which ranged from about 46% in the variant containing a filtered solution of sodium alginate and guar gum to about 60% in the Alg:Chit:Gum variant. In the microspheres composed of unfiltered ternary coating, the content of procyanidins was determined at 98.03 mg/100 g constituting about 62% of the total polyphenol content, while anthocyanins were found at about 20% (31.59 mg/100 g), phenolic acids at 11% (17.17 mg/100 g product), flavonols at 4% (6.31 mg/100 g product), and flavon-3-ols in the form of monomers and dimers at about 3% (4.98 mg/100 g product).

#### 3.1.3. Microcapsules after Storage

Among the stored microcapsules composed of chokeberry concentrate, those made of guar gum were characterized by the highest content of polyphenolic compounds, while the lowest value was recorded for those composed of filtered sodium alginate solution. In the case of the two microsphere variants containing guar gum (without filtration), the total content of polyphenols decreased sharply after two weeks of storage and then increased after another 14 days. In contrast, unfiltered (Alg) and (Alg:Chit) microspheres had an increased amount of polyphenolic compounds during the first two weeks of storage, and then the amount of compounds decreased. The increase in the concentration of polyphenols after the storage time was associated with an increase in the content of proanthocyanidin polymers. It is supposed that due to the degradation of the microspheres over time, there was a gradual release of previously unavailable compounds. It is highly probable that depolymerization occurred during storage. As a result of decomposition, a fraction of encapsulated substance might have been released and combined with the polysaccharide material, resulting in the formation of stable complexes, which contributed to the total polyphenol content. It was found that the best option to preserve anthocyanins was the Alg:Gum coating. After four weeks of storage, just 2% decrease in the total anthocyanin content was observed in the case of unfiltered microspheres.

Similarly to encapsulated chokeberry concentrate, among capsules containing powder, the highest content of polyphenolic compounds was noted in the microspheres composed of guar gum. In the three-component coating variant (Alg:Chit:Gum), the polyphenol content was 166.39 mg/100 g, while in the Alg:Gum variant it was 135.86 mg/100 g product. After two weeks of storage, a sharp decrease in polyphenolic content was observed in both the capsule variants, amounting to about 70% and 35% of the initial determined quantity, respectively (51.25 mg and 85.01 mg/100 g product). After another two weeks of storage, the content of polyphenols in microcapsules made of sodium alginate, chitosan, and guar gum was determined at 144.49 mg, while in microspheres composed of alginate and guar gum the content was 151.43 mg/100 g product. The use of ternary microspheres ensured the preservation of polyphenols at about 87%. This was much higher than the level determined in the encapsulated aronia concentrate, which indicates a better interaction of the coating material with the aronia powder resulting in stabilization of the bioactive compounds. In the Alg:Gum variant, the amounts after storage were higher than the original values. This is probably due to the depolymerization of compounds and their binding with the coating material, forming stable complexes. In the case of the Alg:Chit variant, the content of polyphenolic compounds increased from 56.69 to 115.80 mg/100 g product after the first storage period, while it decreased to 57.65 mg/100 g during the second storage period. Similarly to Alg:Gum (BF), the final content was slightly higher than the initial value. A similar increasing trend was observed for the microspheres composed solely of sodium alginate (Alg), in which the polyphenol content increased from 101.66 to 127.72 mg/100 g product after four weeks of storage.

#### 3.1.4. Polyphenolic Compounds in Chokeberry Concentrate and Chokeberry Powder

The content of polyphenolic compounds indicated above was significantly different from that determined in pure chokeberry concentrate. However, the most similar profile of polyphenolic compounds was observed in the microspheres of ternary composition and those containing guar gum and alginate (without filtration). The total content of phytochemicals in chokeberry concentrate, which was the base for microencapsulation, was 13.62 g/100 g product (Table 2). A much lower concentration of polyphenols amounting to 63.58 g compounds/kg of chokeberry extracts was determined by Bonarska-Kujawa et al. [39] in their study. In the present study, procyanidin polymers were the most abundant group of polyphenols in the concentrate, and constituted 75% (5197.38 mg/100 g) of the total bioactive compounds. A slightly smaller share was represented by anthocyanins. Their content was recorded at a level of about 12% (812.71 mg/100 g), which is lower compared with the results obtained by Zheng and Wang [40], who determined the content of polyphenolic compounds and antioxidant activity in the acetone extract of chokeberry. In the present study, five compounds were identified among the anthocyanins: cyanidin 3-O-galactoside (about 70% of the total anthocyanin content), cyanidin 3-O-arabinoside (23%), cyanidin 3-O-xyloside (about 4%), cyanidin 3-O-glucoside (3%), and pelargonidin 3-O-arabinoside (>1%). The mentioned values of individual anthocyanin compounds are similar to the data presented by Dembczynski et al. [41] who indicated that cyanidin galactoside was the most abundant anthocyanin compound with a level of about 65%. On the other hand, the authors ranked cyanidin arabinoside as the second abundant, which is consistent with the results obtained in this study. All the above mentioned anthocyanins, except pelargonidin arabinoside, were present in the unstored microspheres. Another group of compounds present in chokeberry extract was phenolic acids (approximately 8%), which included chlorogenic acid, neochlorogenic acid, cryptochlorogenic acid, and coumaroylquinic acid. Chlorogenic acid and neochlorogenic acid were the major phenolic acids in chokeberry extract, accounting for approximately 52% and 47%, respectively. A similar level of the main phenolic acids was determined by Oszmiański and Wojdyło [42]. The fourth group of compounds that were found to be abundant in terms of the total polyphenol content in the present study was flavonols (approximately 4%). Seven compounds were identified in this group: quercetin 3-O-galactoside (about 46% of the total number of flavonols), quercetin 3-O-rutinoside (11%), isoramnetin pentosylhexoside (about 11%), quercetin 3-O-glucoside (10%), quercetin 3-O-robinoside (9%), quercetin 3-O-xyanoside (8%), and isoramnetin rhamnosylhexoside (about 5%). A similar finding was reported by Oszmiański and Lachowicz [14]. Only five quercetin derivatives were identified in microcapsules in the present study. The least numerous group of compounds found in the study was mono- and dimers of flavan-3-ols, the total content of which accounted for only 2% of all polyphenolic compounds.

The total polyphenol content of aronia powder was 45.44 g/100 g. This value was significantly higher than the value estimated for powder from freeze-dried fresh fruit (24.72 g/100 g) and powder from the pomace of uncrushed fruit (24.45 g/100 g) in the study by Oszmiański and Lachowicz [14]. A lower amount of polyphenols in freeze-dried chokeberry extract (27.63 g/100 g) was also determined by Horszwald et al. [43]. In the present study, the main group of compounds in chokeberry powder was anthocyanins, with a concentration of 16.45 g/100 g of the product, which corresponded to about 36% of the total amount of polyphenols determined. As in the case of aronia concentrate, five anthocyanins were identified in the powder: cyanidin 3-O-galactoside (66.5% of total anthocyanins), cyanidin 3-O-arabinoside (23%), cyanidin 3-O-xyloside (5.5%), cyanidin 3-O-glucoside (4%), and pelargonidin 3-O-arabinoside (1%). Of these, pelargonidin arabinoside is detected very rarely and represents a minor amount of anthocyanins Sidor and Gramza-Michalowska [44]. The trend of occurrence of the other four anthocyanins was consistent with the results obtained by Oszmiański and Wojdyło [42]. The values obtained for the powder were similar to those obtained for the aronia concentrate. The second largest group of compounds found in chokeberry was polymers of proanthocyanidins, the content of which was determined at 34% (15.10 g/100 g product). This value is lower in comparison with the amount of procyanidins reported by Oszmiański and Lachowicz [14], which was about 40% of the total polyphenol content. The third most abundant group of polyphenols was phenolic acids, the total amount of which was 6.88 g/100 g product, constituting about 15% of the total phytochemicals. 

Similarly to chokeberry concentrate, chlorogenic acid and neochlorogenic acid were found to be the major phenolic acids in the powder, with a proportion of 46% and 54%, respectively. This trend of occurrence is contrary to that of phenolic acids determined for the concentrate. Coumaroylquinic acid and cryptochlorogenic acid were found in trace amounts. Another group of polyphenols that were abundant in the aronia powder was flavonols represented by seven compounds: quercetin 3-O-xyanoside, quercetin 3-O-robinoside, quercetin 3-O-rutinoside, quercetin 3-O-galactoside, quercetin-3-O-glucoside, isoramnetin pentosylhexide, and isoramnetin rhamnosyl-hexoside. Their total content was estimated at 4.96 g/100 g, which corresponded to about 11% of the total amount of polyphenols. Of these, the highest content was found for quercetin galactoside (about 43% of flavonols), and the lowest for isoramnetin hexoside isomer (4.5%). The last group of polyphenols identified in the chokeberry powder was flavan-3-ols in the form of monomers and dimers, which accounted for about 4% of the total polyphenol content (573.50 mg/100 g). Five flavan-3-ols were identified in the powder, namely, glucuronic eriodictyol, procyanidin B2, (+)-catechin, (−)-epicatechin, and PA-trimer type A, the proportions of which were estimated at 43%,15%, 4%, 15%, and 23%, respectively.

Similarly to the encapsulation of chokeberry concentrate, the use of alginate coating combined with guar gum (Alg:Gum) allowed almost complete protection of anthocyanins in chokeberry powder. The concentration of anthocyanins in the nonstored microspheres was decreased by about 8% compared with the stored microspheres. Similar observations on the effect of guar gum on anthocyanin stability were reported by Pieczykolan and Kurek [45]. The highest stability of phenolic acids over time was observed in the coating variant composed of sodium alginate alone, which was not subjected to prior filtration. This microcapsule allowed the preservation of approximately 40% of the initial amount of phenolic acids.

### 3.2. Determinations of Antioxidants Capacity by ABTS, FRAP and ORAC Methods

Chokeberry concentrate had higher ABTS, FRAP and ORAC antioxidant activity values compared to chokeberry powder (Table 2). The highest antioxidant activity of the concentrated extract was obtained by ORAC method (500.57 mmol TE/100 g product). Slightly lower value was recorded for ABTS method (357.62 mmol TE/100 g product) and the lowest value was recorded for FRAP method (254.16 mmol TE/100 g). The results of antioxidant activity measured by ABTS and FRAP method by Oszmiański and Lachowicz [14] in juice from crushed chokeberry fruits, are much lower than the results obtained in this experiment. The mentioned authors obtained 32.73 mmol TE/100 g dry mass and 20.20 mmol Trolox/100 g dry mass, respectively. Lower potency to scavenge the cation radical ABTS was also shown by Oszmiański and Wojdyło [42] in chokeberry juice (314.05 µmol TE/100 g dry mass). However, these differences are due to the purity of the tested materials. In the present study, this parameter was measured for the concentrate of isolated compounds from chokeberry, which was a much purer form than juice or crushed fruit. On the other hand, the highest value of antioxidant activity for the powder was recorded for the method using the ability to adsorb free radicals-ORAC (405.45 mmol TE/100 g product). The strength of ABTS cation radical scavenging capacity was more than two times lower (209.60 mmol TE/100 g of product) in comparison with ORAC method. The lowest value of antioxidant activity was obtained for FRAP method (169.06 mmol TE/100 g of product). A slightly higher antioxidant activity value measured by the FRAP method in chokeberry powder was obtained by Horszwald et al. [43], which was 193.69 mmol TE/100 g dry mass. However, the mentioned author in his study obtained lower ABTS capacity (180.45 µmol TE/100 g dry mass). Additionally, Denev et al. [3] obtained lower antioxidant activity by ORAC method (55,507.7 µmol TE/l) compare to our study.

Table 3 presents the results for the analysis of antioxidant activity of the obtained microspheres by ABTS, FRAP, and ORAC methods. Among the unfiltered chokeberry concentrate capsules, the highest ABTS and FRAP activities were exhibited by the unfiltered Alg:Chit:Gum variant (0.712 and 0.655 mmol TE/100 g product, respectively), while the lowest activity was observed for the samples composed of unfiltered sodium alginate solution (0.309 and 0.242 mmol TE/100 g product). In the ORAC method, the estimated antioxidant activity ranged from 2.398 TE/100 g for the Alg:Gum variant to 1.433 TE/100 g for the single-component alginate capsule. After two weeks of storage, the analysis of antioxidant activity by all three methods showed the highest activities for Alg:Gum capsules. After another two weeks of storage, the highest ABTS radical cation scavenging potential was observed for the ternary microspheres (0.607 TE/100 g). The FRAP method showed the highest activity for Alg:Gum coating (0.595 TE/100 g). The antioxidant activity determined by the ORAC method after storage ranged from 0.699 TE/100 g for the alginate capsules to 1.308 TE/100 g for the Alg:Chit variant. When individual alginate capsules were compared, it was noted that the addition of chitosan or guar gum increased the antioxidant activity in all samples. The use of guar gum resulted in the highest antioxidant activity in the encapsulated chokeberry concentrate. It also contributed to the highest activity after storage. Filtration process significantly decreased the antioxidant activity of the tested microspheres. The highest decrease of ABTS scavenging potential was recorded for Alg:Gum variant (loss of about 55% of initial activity), while the lowest decrease (about 30%) was noted for the capsule consisting of sodium alginate.

In the case of microcapsules containing chokeberry powder, the antioxidant activity measured by the ABTS method ranged from 0.202 TE/100 g for the filtered ternary variant to 0.840 TE/100 g for the variant composed of three polymers (Alg:Chit:Gum). The filtered ternary capsules showed the lowest iron ion-reducing potential (FRAP) (0.151 TE/100 g), while Alg:Gum showed the highest potential (0.771 TE/100 g). The analysis using the ORAC method indicated that the antioxidant activity ranged from 1.340 TE/100 g (Alg:Chit after filtration) to 2.100 TE/100 g (Alg:Chit:Gum without filtration). Due to the presence of chitosan in the alginate coating, unfiltered microspheres showed lower antioxidant activity in ABTS, FRAP, and ORAC analyses, compared to the single-component alginate variant. However, chitosan combined with guar gum was found to be one of the most favorable coating options. Similarly to the capsules containing chokeberry concentrate, the filtration process significantly reduced the antioxidant activity of microspheres containing powder. For example, the ABTS cation scavenging capacity of the Alg:Gum variant decreased by about 30%, while that of the Alg:Chit:Gum variant decreased by about 75%. After two weeks of storage, the Alg:Gum microspheres exhibited the highest antioxidant properties in ABTS, FRAP, and ORAC assays, while the chitosan variants showed the lowest activities. During the two-week storage period, no statistically significant difference in the ABTS antioxidant activity was found between the different microsphere variants, except for the Alg:Chit coating, which showed the lowest antioxidant activity. A further two weeks of storage resulted in a slight decrease in ABTS activity in all capsule variants. Nevertheless, the samples composed of sodium alginate and guar gum continued to exhibit the highest activity (0.808 mmol TE/100 g product), while the variant composed of alginate and chitosan had the lowest value (0.374 mmol TE/100 g product). In these variants, the decrease in antioxidant activity compared to the original value was about 2% and 35%, respectively. Depending on the coating variant used, after 14 days of storage, a decrease or increase in the FRAP antioxidant activity was noted in microspheres containing chokeberry powder. An increase in iron ion-reducing capacity was found for the microspheres composed only of sodium alginate (Alg) and its mixture with guar gum (Alg:Gum) (0.838 and 0.808 mmol TE/100 g product, respectively). On the other hand, for ternary coating and Alg:Chit samples, the measured activity decreased to 0.729 and 0.465 mmol TE/100 g product, respectively. In the case of microspheres composed of unfiltered solutions of sodium alginate and guar gum (Alg:Gum) and those composed of only sodium alginate (Alg), no change in antioxidant activity was observed after an extended storage time of four weeks. This suggests that both the coating variants completely preserved the initial antioxidant properties. The highest decrease during storage (about 50% of the initial value), was observed for the Alg:Chit material. Among the unstored microspheres, those composed of an unfiltered mixture of sodium alginate, chitosan, and guar gum and Alg:Gum variant showed the highest free radical absorption capacity (ORAC). For the unfiltered variants containing added guar gum in the formulation, no decrease in antioxidant activity was detected during the two-week storage period. After another 14 days of storage, a significant decrease in the antioxidant activity was found for all microspheres in the ORAC analysis. The highest absorption capacity of reactive oxygen species was characteristic of capsules composed exclusively of unfiltered sodium alginate solution (Alg). The antioxidant capacity of this variant was 1.630 mmol TE/100 g product, which was about 82% of the initial amount. The lowest activity was found for the microspheres containing chitosan (0.868 mmol TE/100 g product). A comparison of the antioxidant activity of the capsules containing the powder and concentrate revealed that the powdered chokeberry extract was much better preserved. Similarly to the case of concentrate, the best coating option for chokeberry powder was the mixture of guar gum and sodium alginate, while sodium alginate used alone as a coating material resulted in significantly better microencapsulation of chokeberry powder compared to concentrate.

### 3.3. Analysis of α-Amylase and α-Glucosidase Inhibition Assays

Table 4 shows the results for the analysis of anti-α-amylase and anti-α-glucosidase inhibitory effect, which was determined as IC_50_ (mg/mL). In the case of chokeberry concentrate, the α-amylase-inhibiting effect ranged from 12.06 mg/mL for the capsules composed of unfiltered solution of sodium alginate, guar gum, and chitosan (Alg:Chit:Gum) after two weeks of storage to 165.02 mg/mL for the ternary microspheres after four weeks of storage (after filtration). Among the unstored variants, the lowest inhibiting capacity was noted for the Alg:Chit variant (70.22 mg/mL), which was about 30 times higher than the value determined for the chokeberry concentrate (2.24 mg/mL). A significantly worse α-amylase-inhibiting effect was also observed by Worszynowicz et al. [46] for the aqueous and methanolic extract of freeze-dried chokeberry fruits (13.55 and 10.31 mg/mL, respectively). In the variants made using chitosan, a higher α-amylase-inhibiting ability was observed after two weeks of storage. The values determined for the ternary coating and alginate–chitosan complex variants were 12.06 and 87.77 mg/mL, respectively. After four weeks of storage, the most favorable effect in terms of health (lowest inhibition) was again determined for the capsules composed of a mixture of three polymers. Microspheres constructed from a filtered solution of a plant polysaccharide showed a much lower enzyme-inhibiting capacity. During the first storage period, the values decreased, and then increased after further weeks of storage. A significantly stronger α-amylase-inhibiting effect compared to α-glucosidase inhibition was detected for all variants of chokeberry capsules. By contrast, an inverse relationship was observed by Xue Du and Myracle [47] when they analyzed fermented kefir with added chokeberry juice.

For the microspheres with chokeberry powder, the α-amylase-inhibiting capacity ranged from 48.54 mg/mL (variant composed of filtered sodium alginate and gum after two weeks of storage) to 315.68 mg/mL (unfiltered coating containing sodium alginate and guar gum after four weeks of storage). Nevertheless, the IC_50_ value determined for chokeberry powder was 50 times lower (0.94 mg/mL) than the lowest IC_50_ value estimated for the microspheres. Among unstored capsules, the lowest and the most beneficial effect—from the consumer’s point of view—was noted for the variant composed of the unfiltered solution of sodium alginate and chitosan (67.17 mg/mL). The highest effect was found for the microspheres composed only of sodium alginate (113.04 mg/mL). This confirms the previous speculations regarding the effect of chitosan and guar gum on the reduction of α-amylase-inhibiting capacity, which was noted for microcapsules containing chokeberry concentrate in the present study. The application of the filtration process seemed to be unfavorable for the preservation of antidiabetic activity in all variants, except for the Alg:Gum variant (a decrease in value from 85.49 to 71.13 mg/mL). During the two-week storage period, the IC_50_ value for guar gum-containing microspheres decreased, whereas it increased for the other two variants.

The antidiabetic activity, which is expressed as the ability of microspheres with chokeberry concentrate to inhibit α-glucosidase, ranged from 111.22 mg/mL (ternary microspheres without filtration after two weeks of storage) to 906.28 mg/mL (Alg:Gum variant after filtration and four weeks of storage). Among the unstored samples, the most favorable variant consisted of the microspheres composed of three polysaccharide materials (112.84 mg/mL). On the other hand, the microspheres composed of a single alginate coating (Alg) showed the worst antidiabetic activity. These results suggest that both chitosan and guar gum had a positive effect on the α-glucosidase-inhibiting ability. Similarly to the α-amylase-inhibiting ability, the variants consisting of filtered polymer solutions showed significantly worse α-glucosidase-inhibiting activity compared to the unfiltered microspheres. In comparison, among the variants containing chokeberry powder, the tricomponent Alg:Gum:Chit microspheres showed the best α-glucosidase-inhibiting ability (115.63 mg/mL). The least favorable option was the sample prepared using the filtered solution of sodium alginate and chitosan (930.16 mg/mL), as an eightfold difference in the ability to inhibit α-glucosidase was observed. When considering the effect of storage time on α-glucosidase-inhibiting activity, it was noted that the IC_50_ value of microspheres composed only of sodium alginate increased with time. The highest α-glucosidase-inhibiting potential was observed for Alg:Chit:Gum capsules (without filtration) after one month of storage (108.05 mg/mL), whereas the lowest ability was observed for microspheres containing only sodium alginate as a coating material (185.75 mg/mL).

### 3.4. Color Measurement in the CIE L*a*b System

To determine the changes in the color parameters of microspheres containing chokeberry powder and those containing concentrate during storage, color was measured immediately after formulation and after 14 and 28 days of storage at +4 °C (Table 5). For unstored microspheres containing chokeberry concentrate, the value of parameter L*, which determines the brightness of the product, ranged from 12.83 for the variant composed of the unfiltered solution of sodium alginate and guar gum to 33.09 for their filtered counterpart. On the other hand, in the case of microspheres containing chokeberry powder, the L* values ranged from 11.77 for unfiltered ternary microspheres to 34.89 for the filtered microspheres containing Alg:Chit:Gum. A lower L* value was indicative of darker microspheres. An analysis of unfiltered microspheres for determining the composition of polyphenolic compounds revealed that variants containing higher concentrations of anthocyanins were characterized by lower L* values. Similar observations were reported by Kalisz et al. [48] in their study investigating the effects of the addition of colored fruit juices on color. In the present study, a contrasting result was observed for the Alg:Chit:Gum microspheres, which had the brightest color despite high levels of anthocyanin compounds. After two weeks of storage, the L* values increased for most samples (ternary microspheres were again an exception), but they decreased after another two weeks. A higher value of the qualitative color discriminant indicates that the color of the microspheres was increased after storage. Similar observations concerning the increase of L* at the initial stage of storage and the subsequent decrease at the final stage of storage were stated by Scibisz et al. [49], based on their study on stored blueberry jam.

The value of the a* parameter of microspheres containing concentrate ranged from 8.49 to 23.52 (for microspheres composed of unfiltered alginate solution and a filtered solution of sodium alginate and chitosan, respectively). Among the powder capsules, the a* value ranged from 7.71 for the unfiltered Alg:Chit:Gum variant to 27.06 for the sample composed of the filtered Alg:Chit:Gum solution. A positive a* value corresponds to a red color. The variants containing guar gum and chitosan were found to have higher a* values. Lachowicz et al. [50] found that the addition of chitosan for juice clarification decreased a* values and the anthocyanin content of the sample. This was probably due to the interaction of the polysaccharide with the mentioned group of polyphenolic compounds, which appears to be beneficial for encapsulation. The application of the filtration process also caused higher a* values among unstored microspheres. After storage, microspheres containing chokeberry powder showed stronger redness, except for the Alg:Chit:Gum variant, as evidenced by an increase in the determined a* value. 

The b* parameter, which is responsible for the yellow color, was determined to be 2.39–5.30 for the unstored capsules containing concentrate and 0.29–6.79 for the capsules containing powder. In both cases, the lowest value for unfiltered microspheres was recorded for the Alg:Chit:Gum variant. In contrast, the highest value of parameter b* was found for Alg:Gum or Alg:Chit microspheres. The use of the filtration process increased the yellow color of the capsules. Moreover, due to the presence of chitosan, the obtained capsules appeared more yellowish. As the storage time increased, the value of the b* coordinate increased further.

### 3.5. Optical Microscopy Analysis

The analysis of microscopic images of microspheres containing concentrate (Figure 1) and chokeberry powder (Figure 2) showed that they differed in color, shape, structure, and size. Depending on the variant used, the capsules had a dark purple color (Alg:Chit:Gum and Alg:Chit) or red color (Alg:Gum and Alg). Additionally, the variants containing chitosan in their formulations showed discoloration. The application of the filtration process resulted in microspheres more even in shape, uniform in color, and smooth in structure. The intensity of their color was lower, which was confirmed by the color analysis. After storage, the microspheres decreased in size, to a lesser (Alg:Gum variant) or greater extent (Alg:Chit:Gum). The same relationship was also observed for the capsules containing aronia powder.

## 4. Conclusions

The study confirmed that guar gum is an effective coating material and had a significantly positive effect on the stability of the polyphenolic compounds during storage. In addition, it was shown that both chitosan and guar gum had positive effects on the α-glucosidase-inhibiting ability and antioxidant potential. Moreover, the variants consisting of filtered polymer solutions showed significantly worse inhibiting and antioxidant activities compared to the unfiltered microspheres, which was related to the significantly lower concentrations of bioactive compounds in the filtered samples. Generally, microspheres containing chokeberry extract were found to be a good source of biologically active compounds and exhibit valuable health-promoting properties, such as the ability to inhibit α-amylase and α-glucosidase. The findings suggest that these products will be applicable for functional food, by enriching foods with stabilized polyphenolic compounds, allowing controlled release of the encapsulated material, and by masking taste and odor.

## Figures and Tables

**Figure 1 foods-10-01994-f001:**
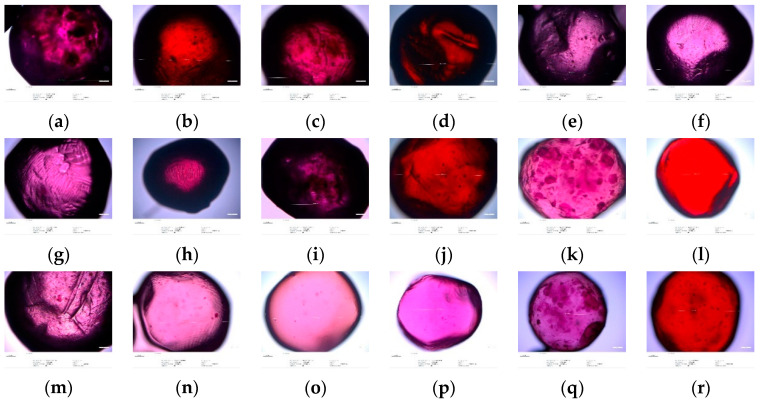
Microscopic images of microspheres coating chokeberry concentrate, made of various coating materials: (**a**) alginate:chitosan:guar gum (non-stored, non-filtered); (**b**) alginate:guar gum (non-stored, non-filtered); (**c**) alginate:chitosan (non-stored, non-filtered); (**d**) alginate (non-stored; non-filtered); (**e**) alginate:chitosan:guar gum (non-stored, after filtrations); (**f**) alginate:guar gum (non-stored, after filtrations); (**g**) alginate:chitosan (non-stored, after filtrations); (**h**) alginate (non-stored, after filtrations); (**i**) alginate:chitosan:guar gum (after 2 weeks of storage, non-filtered); (**j**) alginate:guar gum (after 2 weeks of storage, non-filtered); (**k**) alginate:chitosan (after 2 weeks of storage, non-filtered); (**l**) alginate (after 2 weeks of storage, non-filtered); (**m**) alginate:guar gum (after 2 weeks of storage, after filtrations); (**n**) alginate:guar gum (after 2 weeks of storage, after filtrations); (**o**) alginate:chitosan (after 2 weeks of storage, after filtrations); (**p**) alginate (after 2 weeks of storage, after filtrations); (**q**) alginate:chitosan:guar gum (after 4 weeks of storage, non-filtered); (**r**) alginate:guar gum (after 4 weeks of storage, non-filtered); (**s**) alginate:chitosan (after 4 weeks of storage, non-filtered); (**t**) alginate (after 4 weeks of storage, non-filtered); (**u**) alginate:chitosan:guar gum (after 4 weeks of storage, after filtrations); (**v**) alginate:guar gum (after 4 weeks of storage, after filtrations); (**w**) alginate:chitosan (after 4 weeks of storage, after filtrations); (**x**) alginate (after 4 weeks of storage, after filtrations).

**Figure 2 foods-10-01994-f002:**
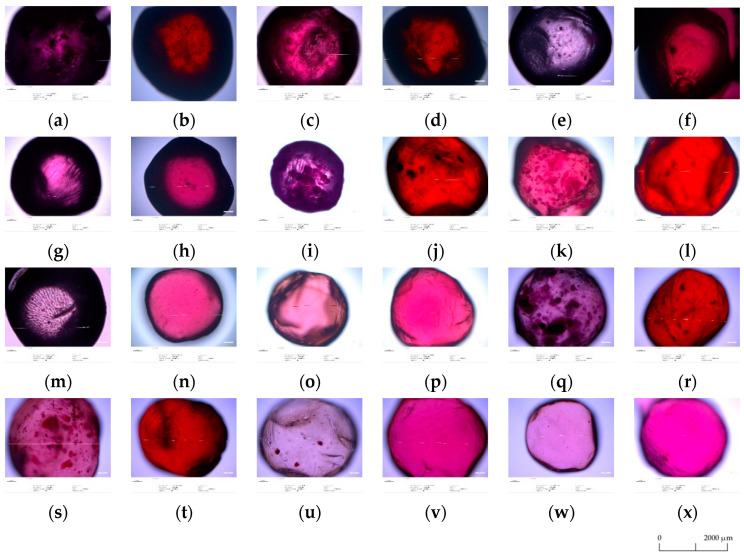
Microscopic images of microspheres coating chokeberry powders, made of various coating materials: (**a**) alginate:chitosan:guar gum (non-stored, non-filtered); (**b**) alginate:guar gum (non-stored, non-filtered); (**c**) alginate:chitosan (non-stored, non-filtered); (**d**) alginate (non-stored; non-filtered); (**e**) alginate:chitosan:guar gum (non-stored, after filtrations); (**f**) alginate:guar gum (non-stored, after filtrations); (**g**) alginate:chitosan (non-stored, after filtrations); (**h**) alginate (non-stored, after filtrations); (**i**) alginate:chitosan:guar gum (after 2 weeks of storage, non-filtered); (**j**) alginate:guar gum (after 2 weeks of storage, non-filtered); (**k**) alginate:chitosan (after 2 weeks of storage, non-filtered); (**l**) alginate (after 2 weeks of storage, non-filtered); (**m**) alginate:guar gum (after 2 weeks of storage, after filtrations); (**n**) alginate:guar gum (after 2 weeks of storage, after filtrations); (**o**) alginate:chitosan (after 2 weeks of storage, after filtrations); (**p**) alginate (after 2 weeks of storage, after filtrations); (**q**) alginate:chitosan:guar gum (after 4 weeks of storage, non-filtered); (**r**) alginate:guar gum (after 4 weeks of storage, non-filtered); (**s**) alginate:chitosan (after 4 weeks of storage, non-filtered); (**t**) alginate (after 4 weeks of storage, non-filtered); (**u**) alginate:chitosan:guar gum (after 4 weeks of storage, after filtrations); (**v**) alginate:guar gum (after 4 weeks of storage, after filtrations); (**w**) alginate:chitosan (after 4 weeks of storage, after filtrations); (**x**) alginate (after 4 weeks of storage, after filtrations).

**Table 1 foods-10-01994-t001:** Polyphenolic compounds of chokeberry microspheres (mg/100 g of products).

Polyphenolic Compounds (mg/100 g of Products)	Storage Time	Types of Microspheres															
		Microspheres with Chokeberry Concentrate								Microspheres with Chokeberry Powder							
		Alg: Chit: Gum (NF/K)	Alg: Gum (NF/K)	Alg: Chit: (NF/K)	Alg (NF/K)	Alg: Chit: Gum (F/K)	Alg: Gum (F/K)	Alg: Chit: (F/K)	Alg (F/K)	Alg: Chit: Gum (NF/P)	Alg: Gum (NF/P)	Alg: Chit: (NF/P)	Alg (NF/P)	Alg: Chit: Gum (F/P)	Alg: Gum (F/P)	Alg: Chit: (F/P)	Alg (F/P)
Total Anthocyanins	Zero times (non stored)	32.90 a	30.58 b	29.53 b	25.89 c	10.65 g	7.64 h	12.19 f	11.52 fg	31.59 C	38.90 AB	21.58 D	41.47 A	5.00 I	11.53 F	9.53 FG	10.17 FG
	After 2 weeks of storage	22.29 cd	30.31 b	15.88 e	25.70 c	6.68 hi	6.00 hi	10.68 g	7.98 h	16.85 E	37.26 B	15.07 E	33.61 BC	2.83 J	8.72 G	6.37 H	8.81 G
	After 4 weeks of storage	20.47 d	30.20 b	15.59 e	21.07 d	1.93 j	5.26 i	10.32 g	7.34 h	12.31 F	35.84 B	7.29 GH	32.29 C	0.00 L	1.44 K	6.29 H	5.57 H
Total Phenolic acid	Zero times (non stored)	12.06 ab	12.93 a	11.59 bc	12.32 a	7.10 e	9.50 cd	6.22 f	10.48 c	17.17 B	17.76 B	10.32 E	22.84 A	4.18 G	17.97 B	7.63 F	13.70 CD
	After 2 weeks of storage	10.54 c	12.26 a	8.10 de	12.68 a	4.09 h	8.79 d	6.35 f	8.98 d	12.94 D	15.63 C	8.06 F	22.21 A	2.98 H	15.24 C	6.65 G	13.15 D
	After 4 weeks of storage	4.96 g	3.26 h	1.33 j	1.87 i	0.00 m	0.75 l	0.96 k	8.59 d	6.64 G	2.92 H	1.24 K	10.78 E	0.00 L	2.23 I	1.38 J	11.44 DE
Total Flavonols	Zero times (non stored)	4.99 a	4.60 ab	5.17 a	4.14 b	2.57 de	2.41 e	2.70 d	2.91 d	6.31 BC	5.86 C	5.02 D	7.58 A	1.22 J	5.17 D	3.23 G	3.99 F
	After 2 weeks of storage	3.95 b	3.83 bc	3.44 c	3.74 bc	0.93 hi	1.61 f	1.54 f	2.20 e	4.39 E	5.03 D	3.55 G	6.67 B	0.19 K	4.05 EF	2.44 I	3.65 FG
	After 4 weeks of storage	2.68 de	1.19 g	0.37 j	0.90 i	0.00 k	0.00 k	1.10 h	1.90 ef	1.53 J	3.01 H	1.11 J	2.44 I	0.00 L	1.54 J	0.80 J	2.74 HI
Total Flavan-3-ols (monomers & dimers)	Zero times (non stored)	4.20 b	3.58 c	4.34 b	4.26 b	2.07 f	2.79 d	2.14 ef	3.13 cd	4.98 B	4.49 B	3.30 D	7.39 A	3.29 DE	4.02 C	2.65 E	3.76 CD
	After 2 weeks of storage	3.49 c	2.55 de	3.77 c	4.09 b	1.33 i	2.46 e	1.76 g	2.84 d	4.16 C	3.83 C	2.23 F	6.03 AB	0.56 H	3.23 DE	2.20 F	3.06 E
	After 4 weeks of storage	5.00 a	2.53 de	3.18 cd	3.59 c	0.30 j	2.29 e	1.61 h	1.78 g	3.49 D	3.40 D	1.52 G	4.92 B	0.34 H	3.11 E	2.08 FG	2.97 E
Total Flavan-3-ols (Procyanidin polymers)	Zero times (non stored)	54.14 a	50.44 b	10.25 h	6.85 i	24.43 e	25.36 e	28.39 de	3.94 j	98.03 B	58.04 E	8.79 O	11.81 N	24.28 J	50.63 F	28.96 I	4.58 S
	After 2 weeks of storage	9.76 h	17.83 fg	49.85 b	32.42 d	4.14 j	3.27 k	28.90 de	29.46 d	7.30 P	16.69 L	81.63 C	72.13 D	2.13 T	6.31 R	29.73 I	31.53 HI
	After 4 weeks of storage	39.70 c	54.28 a	45.72 b	30.27 d	15.53 g	19.85 f	24.29 e	30.89 d	12.97 M	106.01 A	46.83 G	80.89 C	16.59 L	34.16 H	31.01 I	22.21 K
Total content of polyphenols	Zero times (non stored)	108.29 a	102.13 b	60.88 g	53.46 h	46.82 j	47.70 ij	51.64 hi	31.98 l	158.08 A	125.05 D	49.01 I	91.09 F	37.97 KL	89.32 F	52.00 I	36.20 L
	After 2 weeks of storage	50.03 i	66.78 f	81.04 d	78.63 d	17.17 p	22.13 n	49.23 i	51.46 i	45.64 J	78.44 G	110.54 E	140.65 C	8.69 N	37.55 L	47.39 IJ	60.20 H
	After 4 weeks of storage	72.81 e	91.46 c	66.19 f	57.70 g	17.76 o	28.15 m	38.28 k	50.50 i	146.94 B	151.18 AB	57.99 HI	131.32 D	16.93 M	42.48 K	41.56 K	44.93 J

Alg:Chit:Gum—Alginate:Chitosan: Guar gum; Alg:Gum—Alginate:Guar gum; Alg:Chit—Alginate:Chitosan; Alg—Alginate; NF—No filtration; F—Filtration; K—chokeberry concentrate; P—chokeberry powder. Letters (a–p) indicate the significant difference between individual results of total anthocyanins/total phenolic acids/total flavonols/total flavan-3-ols (monomers and dimers)/total polymeric proanthocyanidins)/total polyphenols of all microspheres containing chokeberry concentrate (stored and unstored capsules), according to Duncan’s test. *p* < 0.05. Letters (A–O) indicate the significant difference between individual results of total anthocyanins/total phenolic acids/total flavonols/total flavan-3-ols (monomers and dimers)/total polymeric proanthocyanidins)/total polyphenols of all microspheres containing chokeberry powder (stored and unstored capsules), according to Duncan’s test. *p* < 0.05.

**Table 2 foods-10-01994-t002:** Polyphenolic compounds (mg/100 g) and antioxidant activity (mmol TE/100 g) and anti- α-amylase and anti- α-glucosidase activities (IC_50_) of chokeberry products.

Components and Properties				Chokeberry Powder	Chokeberry Concentrate
Polyphenolic Compounds	Retention Time (R_t_)	[H-M]^−^ (*m*/*z*)	MS/MS (*m*/*z*)	[mg/100 g]	[mg/100 g]
Cyanidin-3-O-galactoside	4.27	449^+^	287	10,940.10 ± 121.45 b	2836.88 ± 52.73 *b*
Cyanidin-3-O-glucoside	4.48	449^+^	287	647.42 ± 6.84 i	134.83 ± 4.28 *i*
Cyanidin-3-O-arabinoside	4.68	419^+^	287	3823.07 ± 46.46 c	918.34 ± 20.07 *e*
Cyanidin-3-O-xyloside	4.99	419^+^	287	900.20 ± 21.61 g	156.57 ± 2.11 *h*
Pelargonidin-3-O-arabinoside	5.17	403^+^	271	143.82 ± 0.39 p	16.89 ± 0.59 *r*
Σ Anthocyanins				16,454.46 ± 196.75 A	4063.51 ± 79.78 *B*
Neochlorogenic acid	3.45	353	191	3697.23 ± 86.32 d	1287.54 ± 19.36 *d*
3-O-p-Coumaroylquinic acid	3.61	337	191	3.39 ± 0.92 t	0.68 ± 0.04 *t*
Chlorogenic acid	4.13	353	191	3169.39 ± 59.06 e	1478.55 ± 37.11 *c*
Cryptochlorogenic acid	4.18	353	191	14.39 ± 1.43 s	2.99 ± 0.01 *s*
Σ Phenolic acid				6883.78 ± 147.73 C	2769.76 ± 56.52 *C*
Quercetin-3-O-vicianoside	5.80	595	432/301	462.60 ± 2.93 k	84.47 ± 1.93 *n*
Quercetin-3-O-robinobioside	6.07	609	463/301	498.91 ± 15.30 j	86.11 ± 1.36 *n*
Quercetin-3-rutinoside	6.14	609	463/301	688.17 ± 16.17 h	115.34 ± 0.95 *j*
Quercetin-3-galactoside	6.30	463	301	2147.48 ± 39.70 f	462.81 ± 3.75 *f*
Quercetin-3-glucoside	6.39	463	301	500.31 ± 11.78 j	100.27 ± 2.05 *l*
Isorhamnetin pentosylhexoside	6.76	609	315	437.63 ± 12.12 l	108.81 ± 0.52 *k*
Isorhamnetin rhamnosyl hexoside isomer	6.83	623	463/315	225.98 ± 2.72 o	53.59 ± 0.72 *o*
Σ Flavonols				4961.04 ± 100.72 D	1011.38 ± 10.28 *D*
(+)-catechin	4.26	289	289	62.38 ± 3.29 r	41.55 ± 0.11 *p*
Procyanidin B2	4.48	577		234.34 ± 6.25 n	41.55 ± 0.84 *p*
(−)-epicatechin	4.56	289	577/289	230.51 ± 4.35 no	93.33 ± 0.35 *m*
A-type PA-trimer	5.01	866	287	350.62 ± 4.16 m	100.59 ± 3.49 *l*
Eriodictynol-glucuronide	6.28	463		655.72 ± 6.46 i	296.49 ± 3.03 *g*
Σ Flawan-3-ols (mono and dimers)				1533.57 ± 24.51 E	573.50 ± 7.82 *E*
Procyanidin polymers				15,607.24 ± 99.32 a B	5197.38 ± 48.73 a *A*
Antioxidant activity				[mmol TE/100 g]	[mmol TE/100 g]
ABTS				357.62 ± 9.023	209.60 ± 6.718
FRAP				254.16 ± 1.189	169.06 ± 4.683
ORAC				500.60 ± 27.880	405.45 ± 16.255
Enzyme inhibitory activity α-amylase α-glucosidase				[mg/mL] 0.94 ± 0.01 0.29 ± 0.04	[mg/mL] 2.24 ± 0.04 1.14 ± 0.08

[H-M]^−^—deprotonated pseudo molecular ions; [H+M]^−^—protonated pseudo molecular ions; MS/MS—ion fragmentation in mass spectrometry; letters (a–t) indicate the significant difference between individual results of phenolic compounds identified in the chokeberry powder, in turn letters (A–E) present differences between each fraction determined in chokeberry powder, both of them according to Duncan’s test. *p* ≤ 0.05. Letters (*a–t*) indicate the significant difference between individual results of phenolic compounds identified in the chokeberry concentrate, in turn letters (*A*–*E*) present differences between each fraction determined in chokeberry concentrate, both of them according to Duncan’s test. *p* ≤ 0.05.

**Table 3 foods-10-01994-t003:** Antioxidant activity (mmol TE/100 g) of microspheres with chokeberry concentrate and chokeberry powder.

Types of Microspheres	Zero Time (Non Stored)			Time after 2 Weeks of Storage			Time after 4 Weeks of Storage		
	ABTS (mmol TE/100 g)	FRAP (mmol TE/100 g)	ORAC (mmol TE/100 g)	ABTS (mmol TE/100 g)	FRAP (mmol TE/100 g)	ORAC (mmol TE/100 g)	ABTS (mmol TE/100 g)	FRAP (mmol TE/100 g)	ORAC (mmol TE/100 g)
Alg:Chit:Gum (NF/K)	0.712 ± 0.045 ab	0.655 ± 0.013 b	2.038 ± 0.055 c	0.677 ± 0.050 b	0.650 ± 0.005 b	2.042 ± 0.031 c	0.607 ± 0.004 c	0.542 ± 0.008 d	1.240 ± 0.048 kl
Alg:Gum (NF/K)	0.713 ± 0.038 ab	0.652 ± 0.016 b	2.398 ± 0.001 b	0.729 ± 0.009 a	0.678 ± 0.005 a	2.628 ± 0.038 a	0.601 ± 0.008 c	0.595 ± 0.000 c	1.229 ± 0.064 l
Alg:Chit (NF/K)	0.570 ± 0.019 cd	0.542 ± 0.003 d	1.857 ± 0.060 d	0.567 ± 0.017 cd	0.474 ± 0.015 e	1.322 ± 0.053 j	0.588 ± 0.016 c	0.460 ± 0.003 e	1.308 ± 0.071 kj
Alg (NF/K)	0.446 ± 0.021 f	0.402 ± 0.018 g	1.433 ± 0.008 hi	0.541 ± 0.042 cd	0.463 ± 0.031 e	1.378 ± 0.047 ij	0.522 ± 0.008 e	0.434 ± 0.005 f	0.997 ± 0.025 m
Alg: Chit: Gum (F/K)	0.324 ± 0.006 h	0.254 ± 0.021 ij	1.729 ± 0.023 e	0.267 ± 0.026 cd	0.207 ± 0.003 k	1.346 ± 0.034 j	0.277 ± 0.008 ij	0.188 ± 0.018 k	0.867 ± 0.022 n
Alg: Gum (F/K)	0.315 ± 0.021 hi	0.243 ± 0.004 ij	1.627 ± 0.059 f	0.315 ± 0.009 j	0.244 ± 0.016 ij	1.515 ± 0.064 g	0.310 ± 0.021 hi	0.235 ± 0.002 ij	0.710 ± 0.037 o
Alg: Chit (F/K)	0.332 ± 0.024 h	0.261 ± 0.023 i	1.511 ± 0.030 g	0.280 ± 0.025 hij	0.231 ± 0.019 j	1.432 ± 0.020 hi	0.304 ± 0.009 hij	0.243 ± 0.019 ij	0.732 ± 0.053 o
Alg (F/K)	0.309 ± 0.021 hi	0.242 ± 0.013 ij	1.464 ± 0.013 gh	0.383 ± 0.003 ij	0.310 ± 0.011 h	1.030 ± 0.055 m	0.403 ± 0.007 g	0.324 ± 0.008 h	0.699 ± 0.019 o
Alg:Chit:Gum (NF/P)	0.840 ± 0.043 AB	0.763 ± 0.005 B	2.100 ± 0.068 AB	0.811 ± 0.014 BC	0.729 ± 0.030 C	2.026 ± 0.021 BC	0.733 ± 0.021 D	0.609 ± 0.010 D	1.172 ± 0.024 IJ
Alg:Gum (NF/P)	0.823 ± 0.031 AB	0.771 ± 0.023 B	2.083 ± 0.073 AB	0.870 ± 0.048 A	0.808 ± 0.016 A	2.154 ± 0.028 A	0.808 ± 0.039 BC	0.814 ± 0.001 A	1.283 ± 0.060 GH
Alg:Chit (NF/P)	0.578 ± 0.011 E	0.540 ± 0.007 E	1.854 ± 0.017 D	0.582 ± 0.017 E	0.465 ± 0.020 F	1.124 ± 0.053 JK	0.374 ± 0.004 HI	0.273 ± 0.004 KL	0.868 ± 0.057 M
Alg (NF/P)	0.809 ± 0.032 BC	0.763 ± 0.001 B	1.989 ± 0.066 C	0.829 ± 0.027 AB	0.838 ± 0.003 A	1.362 ± 0.044 G	0.773 ± 0.004 CD	0.823 ± 0.000 A	1.630 ± 0.137 E
Alg: Chit: Gum (F/P)	0.202 ± 0.010 LM	0.151 ± 0.009 O	1.457 ± 0.077 F	0.219 ± 0.005 L	0.163 ± 0.003 NO	1.237 ± 0.045 HI	0.170 ± 0.008 M	0.163 ± 0.003 NO	0.641 ± 0.033 O
Alg: Gum (F/P)	0.565 ± 0.030 E	0.428 ± 0.037 G	1.868 ± 0.024 D	0.462 ± 0.015 G	0.362 ± 0.013 I	2.056 ± 0.040 BC	0.513 ± 0.004 F	0.396 ± 0.009 H	1.055 ± 0.031 KL
Alg: Chit (F/P)	0.320 ± 0.019 JK	0.244 ± 0.026 LM	1.340 ± 0.041 G	0.325 ± 0.023 JK	0.253 ± 0.008 LM	0.847 ± 0.032 M	0.302 ± 0.010 K	0.232 ± 0.023 M	0.725 ± 0.019 NO
Alg (F/P)	0.337 ± 0.067 IJK	0.297 ± 0.039 JK	1.579 ± 0.002 E	0.363 ± 0.011 IJ	0.284 ± 0.018 K	0.987 ± 0.033 L	0.417 ± 0.006 H	0.322 ± 0.010 J	0.749 ± 0.034 N

Alg:Chit:Gum—Alginate:Chitosan: Guar gum; Alg:Gum—Alginate:Guar gum; Alg:Chit—Alginate:Chitosan; Alg—Alginate; NF—No filtration; F—Filtration; K—chokeberry concentrate; P—chokeberry powder. Letters (a–o) indicate the significant difference between individual results of antioxidant activity measured by a specific method of all microspheres containing chokeberry concentrate (stored and unstored capsules), according to Duncan’s test. *p* < 0.05. Letters (A–O) indicate the significant difference between individual results of antioxidant activity measured by a specific method of all microspheres containing chokeberry powder (stored and unstored capsules), according to Duncan’s test. *p* < 0.05.

**Table 4 foods-10-01994-t004:** Anti-α-amylase and anti-α-glucosidase activities.

Types of Microspheres		Enzyme Inhibitory Activity of α-Amylase IC_50_ (mg/mL)			Enzyme Inhibitory Activity of α-Glucosidase IC_50_ (mg/mL)		
		Zero Time (Non Stored)	after 2 Weeks of Storage	after 4 Weeks of Storage	Zero Time (Non Stored)	After 2 Weeks of Storage	after 4 Weeks of Storage
Microspheres with chokeberry concentrate (K)	Alg: Chit: Gum (NF)	84.15 ± 2.72 e	12.06 ± 0.11 a	74.96 ± 1.85 c	112.84 ± 2.86 *a*	111.22 ± 2.85 *a*	121.49 ± 2.70 *b*
	Alg: Gum (NF)	70.22 ± 2.81 c	137.01 ± 3.72 ij	93.19 ± 1.17 f	133.34 ± 3.05 *c*	120.06 ± 3.75 *b*	198.94 ± 3.71 *e*
	Alg: Chit (NF)	108.98 ± 0.99 g	87.77 ± 1.63 e	140.51 ± 2.01 j	137.09 ± 2.96 *c*	123.97 ± 1.85 *b*	117.75 ± 3.21 a*b*
	Alginian (NF)	127.63 ± 1.38 h	146.66 ± 2.05 k	154.97 ± 1.97 l	189.68 ± 3.11 *d*	212.30 ± 4.40 *f*	275.81 ± 5.80 *h*
	Alg: Chit: Gum (F)	152.52 ± 2.01 l	81.25 ± 1.53 d	165.02 ± 1.57 m	452.81 ± 10.54 *k*	494.56 ± 7.23 *m*	360.46 ± 7.77 *i*
	Alg: Gum (F)	140.45 ± 1.69 j	23.12 ± 0.07 b	126.89 ± 2.00 h	401.14 ± 7.97 *j*	474.50 ± 8.05 *l*	906.28 ± 17.75
	Alg: Chit (F)	160.37 ± 4.12 lm	154.28 ± 3.11 l	134.45 ± 2.63 i	413.05 ± 8.15 *j*	223.50 ± 3.49 *g*	623.24 ± 19.46 *o*
	Alginian (F)	136.37 ± 1.12 i	78.94 ± 1.95 d	79.55 ± 0.63 d	414.43 ± 9.17 *j*	228.57 ± 5.28 *g*	520.53 ± 10.53 *n*
Microspheres with chokeberry powder (P)	Alg: Chit: Guma (NF)	85.82 ± 1.03 F	76.71 ± 1.27 E	69.80 ± 1.02 CD	115.63 ± 3.13 *C*	113.69 ± 2.00 *C*	108.05 ± 2.18 *B*
	Alg: Guma (NF)	85.49 ± 0.59 F	54.34 ± 0.03 B	315.68 ± 5.93 M	142.94 ± 1.52 *F*	114.75 ± 1.99 *C*	125.23 ± 2.84 *D*
	Alg: Chit (NF)	67.17 ± 1.73 C	102.82 ± 1.53 G	109.04 ± 2.03 H	131.66 ± 2.07 *E*	100.57 ± 2.21 *A*	170.87 ± 1.53 *G*
	Alginian (NF)	113.04 ± 1.93 H	114.36 ± 1.73 H	85.84 ± 0.45 F	122.92 ± 2.39 *D*	131.64 ± 1.73 *E*	185.75 ± 3.05 *H*
	Alg: Chit: Guma (F)	213.94 ± 3.21 K	111.69 ± 1.99 H	127.69 ± 2.74 I	608.30 ± 16.32 *P*	626.79 ± 10.10 *P*	390.51 ± 7.00 *N*
	Alg: Guma (F)	71.13 ± 1.03 D	48.54 ± 0.24 A	130.40 ± 1.06 I	169.33 ± 4.02 *G*	194.97 ± 1.11 *I*	261.62 ± 1.53 *K*
	Alg: Chit (F)	250.23 ± 1.85 L	127.52 ± 1.62 I	87.50 ± 2.63 F	930.16 ± 21.54 *S*	520.50 ± 13.09 *O*	683.48 ± 14.71 *R*
	Alginian (F)	151.46 ± 2.00 J	112.97 ± 2.03 H	216.32 ± 4.01 K	208.76 ± 4.61 *J*	334.25 ± 2.22 *L*	363.32 ± 4.41 *M*

Alg:Chit:Gum—alginate:chitosan:guar gum; Alg:Gum—alginate:guar gum; Alg:Chit—alginate:chitosan; Alg—alginate; NF—no filtration; F—filtration; K—chokeberry concentrate; P—chokeberry powder. Letters indicate significant differences between individual results of inhibitory activity against α-amylase (a–m) and α-glycosidase (*a*–*n*) of all microspheres containing chokeberry concentrate (stored and unstored capsules), according to Duncan’s test, *p* ≤ 0.05. Letters indicate the significant differences between individual results of inhibitory activity against α-amylase (A–M) and α-glycosidase (*A*–*S*) of all microspheres containing chokeberry powder (stored and unstored capsules), according to Duncan’s test, *p* ≤ 0.05.

**Table 5 foods-10-01994-t005:** Color measurement parameters for microspheres.

Types of Microspheres	L*			a*			b*		
	Zero Times (Non Stored)	after 2 Weeks of Storage	after 4 Weeks of Storage	Zero Times (Non Stored)	after 2 Weeks of Storage	after 4 Weeks of Storage	Zero Times (Non Stored)	after 2 Weeks of Storage	after 4 Weeks of Storage
Alg:Chit:Gum (NF/K)	27.16 ± 0.74 h	21.23 ± 0.25 f	15.86 ± 0,68 cd	19.96 ± 0.83 k	16.72 ± 0.25 n	18.19 ± 1.00l m	2.53 ± 0.31 l	2.66 ± 0.68 l	3.39 ± 0.73 kl
Alg:Gum (NF/K)	12.83 ± 0.69 b	15.18 ± 0.38 c	11.27 ± 0.62 a	17.15 ± 1.18 mn	35.15 ± 0.21 c	33.49 ± 0.15 d	5.17 ± 0.35 ih	17.72 ± 0.22 b	15.91 ± 0.34 c
Alg:Chit (NF/K)	16.84 ± 0.70 d	22.97 ± 0.40 g	13.57 ±0.58 b	18.36 ± 1.76 p	20.53 ± 0.70 jk	10.60 ± 0.49 o	3.78 ± 0.40 k	4.05 ± 0.49 jk	1.46 ±0.37 m
Alg (NF/K)	19.38 ± 0.48 e	21.99 ± 0.86 fg	15.29 ± 0.65 c	8.49 ± 2.05 lm	28.36 ± 0.28 f	39.08 ± 0.61 b	3.32 ± 0.52 lk	12.52 ± 0.27 d	23.98 ± 0.67 a
Alg: Chit: Gum (F/K)	29.51 ± 0.59 i	22.94 ± 0.63 g	35.67 ± 0.33 k	21.91 ± 0.8 1ij	8.44 ± 0.96 p	19.17 ± 0.60 kl	4.29± 0.02 ijk	−2.54 ± 0.93 n	11.46 ± 0.95 e
Alg: Gum (F/K)	33.09 ± 0.57 j	47.77 ± 0.96 m	15.91 ± 0.71 k	22.74 ± 0.55 hi	24.24 ± 0.15 g	33.28 ± 1.03 d	5.40 ± 0.21 h	5.96 ± 0.52 h	11.62 ± 1.02 de
Alg: Chit (F/K)	22.61 ± 0,51 g	37.07 ± 0.55 l	28.56 ± 0.38 i	23.52 ± 0.7 gh	30.8 ± 0.61e	41.46 ± 0.48 a	4.99± 0.47 hij	7.58 ± 0.54 g	17.56 ± 1.26 b
Alg (F/K)	21.90 ± 0,65 fg	26.74 ± 0.41 h	21.82 ± 0.93 fg	19.60 ± 0.75 ik	29.37 ± 0.67 f	24.63 ± 0.17 g	2.39 ± 0.21 m	8.63 ± 0.59 f	5.88 ± 0.63 h
Alg:Chit:Gum (NF/P)	11.77 ± 0.55 A	34.43 ± 0.71 G	10.55 ± 0.62 A	7.71 ± 1.44 L	4.88 ± 0.62 N	8.05 ± 0.39 L	0.29 ± 0.55 l	−2.31 ± 0.85 N	−0.76 ± 0.81 M
Alg:Gum (NF/P)	15.46 ± 0.37 B	15.96 ± 0.88 B	12.11 ± 0.79 A	8.23 ± 1.23 L	36.31 ± 0.58 C	18.99 ± 0.46 I	1.79 ± 0.45 k	19.72 ± 0.51 A	6.70 ± 0.11 F
Alg:Chit (NF/P)	19.14 ± 0.14 C	24.97 ± 0.82 E	10.84 ± 0.66 A	21.02 ± 1.59 GH	27.51 ± 0.64 E	14.88 ± 1.08 J	4.12 ± 0.75 m	7.29 ± 0.97 F	2.41 ± 0.96 JK
Alg (NF/P)	12.08 ± 0.83 A	23.76 ± 0.72 DE	10.40 ± 0.34 A	11.05 ± 1.39 K	22.08 ± 0.45 G	22.04 ± 0.76 G	2.96 ± 0.7 1ij	9.52 ± 0.97 E	8.76 ± 0.41 E
Alg: Chit: Gum (F/P)	34.89 ± 0.80 G	60.67 ± 0.66 I	47.68 ± 0.38 H	10.93 ± 0.55 K	13.49 ± 0.81 J	14.72 ± 0.40 J	2.85 ± 0.14 ijk	8.83 ± 0.79 E	13.93 ± 0.68 C
Alg: Gum (F/P)	21.31 ± 0.12 CD	34.98 ± 0.82 G	16.42 ± 0.32 CD	19.01 ± 0.72 I	42.71 ± 0.33 A	39.22 ± 1.17 B	5.39 ± 0.45 g	16.44 ± 0.72 B	20.65 ± 0.58 A
Alg: Chit (F/P)	29.53 ± 0.35 F	45.50 ± 0.57 H	2.41 ± 0.55 F	27.06 ± 0.81 E	25.20 ± 0.25 F	39.50 ± 0.61 B	6.79 ± 0.58 f	7.13 ± 0.55 F	15.58 ± 0.22 B
Alg (F/P)	25.35 ± 0.09 E	33.78 ± 0.76 G	19.30 ± 0.73 C	19.69 ± 0.44 HI	34.32 ± 0.51 D	33.04 ± 0.72 D	3.65 ± 0.86 hi	11.30 ± 0.60 D	11.50 ± 0.59 D

Alg:Chit:Gum—alginate:chitosan:guar gum; Alg:Gum—alginate:guar gum; Alg:Chit—alginate:chitosan; Alg—alginate; NF—no filtration; F—filtration; K—chokeberry concentrate; P—chokeberry powder. Letters (a–o) indicate significant differences between individual results for a given color parameter for all microspheres containing chokeberry concentrate (stored and unstored capsules), according to Duncan’s test. *p* < 0.05. Letters (A–N) indicate significant differences between individual results for a given color parameter for all microspheres containing chokeberry powder (stored and unstored capsules), according to Duncan’s test. *p* < 0.05.

## Data Availability

All the data are reported in the article.

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
