# Peer review of "The Types of Polysaccharide Coatings and Their Mixtures as a Factor Affecting the Stability of Bioactive Compounds and Health-Promoting Properties Expressed as the Ability to Inhibit the α-Amylase and α-Glucosidase of Chokeberry Extracts in the Microencapsulation Process"

_foods, 2021, doi:10.3390/foods10091994_

Round 1

Reviewer 1 Report

  1. This study investigates the use of polysaccharides coating to affect the properties of chokeberry extracts and the resulting data would be useful in the related field. However, some major concerns should be addressed.
  2. Scale bars should be applied to figures 1 & 2.
  3. Statistics should be applied to all data including those in tables 1, 2, 4, 5. Importantly, the data of enzyme inhibitory activities should have a significant difference between means in each treatment. The result and conclusion must be described according to the statistics.
  4. The numbering of tables has a problem that the older is 1, 2, 3, 4 and then 3? I think it should be 5.
  5. The letters that mark the significant differences between means in tables 3 & 5(3?) are difficult to understand, so please add a note for describing their meanings below the table.
  6. In the title, it should be more specific for the test of the inhibitions of α-amylase and α-glucosidase but not just use "health-promoting properties".
  7. There is a lack of a section of "discussion".
  8. It needs iconic spectra of bioactive constituents such as polyphenolic compounds in the control as well as the best treatments.

Author Response

Thank you for your review. Below are presented the answers to your suggestions

The manuscript has been revised according to the reviewers’ suggestions. I used red lettering for the changes made to the revised manuscript.

Reviewer 1:

  1. Scale bars should be applied to figures 1 & 2.

Answer: The scale has been added below the figures. Please see in the revised paper.

  1. Statistics should be applied to all data including those in tables 1, 2, 4, 5. Importantly, the data of enzyme inhibitory activities should have a significant difference between means in each treatment. The result and conclusion must be described according to the statistics.

Answer: The data presented in the tables 1, 2, 4, and 5 have been completed with statistics. Additionally, as suggested by the Reviewer, the results and conclusion have been presented according to the statistics. Please see in the revised paper.

  1. The numbering of tables has a problem that the older is 1, 2, 3, 4 and then 3? I think it should be 5.

Answer: Yes, it was a mistake. This table should be 5. It has been changed. Please see in the revised paper

  1. The letters that mark the significant differences between means in tables 3 & 5(3?) are difficult to understand, so please add a note for describing their meanings below the table.

Answer: Below the table we have included a description that helps to understand the letters used to denote significant differences between the means. Please see in the revised paper.

  1. In the title, it should be more specific for the test of the inhibitions of α-amylase and α-glucosidase but not just use "health-promoting properties".

Answer: The title has been corrected. Please see in the revised paper.

  1. There is a lack of a section of "discussion".

Answer: The discussion of the results has been combined with the "results" section to make the obtained research results more understandable. Due to the scarce information in the literature on chokeberry microspheres, some paragraphs may have a longer or shorter discussion of the results. The paucity of information in the literature regarding the polyphenolic compound content and antioxidant activity of microspheres caused that the discussion of results in some paragraphs is more concerned with the encapsulated material (i.e. aronia concentrate and powder). In Section 3.2 (Determinations of Antioxidants Capacity by ABTS, FRAP and ORAC Methods), the discussion of results is expanded. Please see in the revised paper.

  1. It needs iconic spectra of bioactive constituents such as polyphenolic compounds in the control as well as the best treatments.

Answer: I believe that there is no point in presenting selected chromatograms directly in the manuscript or in its supplement. The content of polyphenolic compounds in chokeberry has been presented many times by our team and other authors, and the profile of this raw material is well recognized. Additional data in the presented manuscript will reduce the readability of the results. The article is sufficiently extensive and composed of a lot of data. In addition, to be fair, a compound profile should be presented for at least 6 samples (chokeberry powder, chokeberry concentrate, capsules based on powder and concentrate with the highest and the lowest concentration of bioactive compounds) at 4 wavelengths (280 nm, 320 nm, 360 nm, 520 nm) giving a total of 24 chromatograms.

However, in order to clarify the results, the retention times and m/z have been added in Table 2. Additionally, in response to the reviews, I present below selected chromatograms for the reviewer's attention.

Reviewer 2 Report

Presented investigation is very interesting and necessary for areas connected with topics of food. A lot of experiments have been performed to investigate and justify effectiveness of developed active microspheres. I have some suggestions to improve the manuscript:

  1. Check whole manuscript: use ºC as unity of temperature. The ´º´ is not seen.
  2. Paragraph 2.5: add exact composition of gradient of mobile phase
  3. Paragraph 2.7: was colorimeter calibrated against white standard before measurement?
  4. Paragraph 2.7: plastic petri dish was placed on surface with which color? White?
  5. Table 3: After table 3 add sentences that different letters a, b, c….. from the table indicates significant differences between samples.
  6. Tables in general: add unities inside of the table close to name of measured properties for example ABTS (mmol TE/100 g).
  7. Why there aren´t SDs as errors in tables 1, 2, and 4? How many replicated were measured in those cases?

Author Response

Thank you for your comments. We are very grateful for the reviews. We are also very pleased that you find our research interesting and useful in food related fields. Please see below and in the corrected manuscript our detailed response to comments.

The manuscript has been revised according to the reviewers’ suggestions. I used red lettering for the changes made to the revised manuscript.

Thank you for your review. Below are presented the answers to your suggestions

Reviewer 2

  1. Check whole manuscript: use ºC as unity of temperature. The ´º´ is not seen.

Answer: It has been corrected. Please see in the revised paper.

  1. Paragraph 2.5: add exact composition of gradient of mobile phase

Answer: It has been added in the revised paper.

  1. Paragraph 2.7: was colorimeter calibrated against white standard before measurement?

Answer: Yes. The preparation of the device for work was in accordance with the instructions of the measuring machine. Calibration was performed for white and black standards.

  1. Paragraph 2.7: plastic petri dish was placed on surface with which color? White?

Answer: It was a colorless petri dish, which was not on the surface of the capsules, but served as a vessel in which they were placed and evenly distributed during the color analysis. The microspheres were not covered by the dish so that the measuring device was in contact with the surface of the microsphere.

  1. Table 3: After table 3 add sentences that different letters a, b, c….. from the table indicates significant differences between samples.

Answer: It has been corrected. Please see in the revised paper.

  1. Tables in general: add unities inside of the table close to name of measured properties for example ABTS (mmol TE/100 g).

Answer: It has been corrected. Please see in the revised paper.

  1. Why there aren´t SDs as errors in tables 1, 2, and 4? How many replicated were measured in those cases?

Answer: All tables show the average results from three replicates of the analysis. Statistics have been completed.

Round 2

Reviewer 1 Report

It has been revised accordingly.